# Essential Oil Chemotypes and Genetic Variability of *Cinnamomum verum* Leaf Samples Commercialized and Cultivated in the Amazon

**DOI:** 10.3390/molecules27217337

**Published:** 2022-10-28

**Authors:** Júlia Karla A. M. Xavier, Talissa Gabriele C. Baia, Oscar Victor C. Alegria, Pablo Luis B. Figueiredo, Adriana R. Carneiro, Edith Cibelle de O. Moreira, José Guilherme S. Maia, William N. Setzer, Joyce Kelly R. da Silva

**Affiliations:** 1Programa de Pós-Graduação em Química, Universidade Federal do Pará, Belém 66075-900, Brazil; 2Programa Institucional de Bolsas de Iniciação Científica, Universidade Federal do Pará, Belém 66075-900, Brazil; 3Centro de Genômica e Biologia de Sistemas, Universidade Federal do Pará, Belém 66075-900, Brazil; 4Departamento de Ciências Naturais, Centro de Ciências Sociais e Educação, Universidade do Estado do Pará, Belém 66050-540, Brazil; 5Instituto de Estudos em Saúde e Biológicas, Universidade Federal do Sul e Sudeste do Pará, Marabá 68501-970, Brazil; 6Programa de Pós-Graduação em Química, Universidade Federal do Maranhão, São Luís 65080-805, Brazil; 7Aromatic Plant Research Center, 230 N 1200 E, Suite 100, Lehi, UT 84043, USA

**Keywords:** Lauraceae, benzenoids, phenylpropanoids, DNA barcode, *psbA*-*trnH*, GC-MS

## Abstract

*Cinnamomum verum* (Lauraceae), also known as “true cinnamon” or “Ceylon cinnamon” has been widely used in traditional folk medicine and cuisine for a long time. The systematics of *C. verum* presents some difficulties due to genetic variation and morphological similarity between other *Cinnamomum* species. The present work aimed to find chemical and molecular markers of *C. verum* samples from the Amazon region of Brazil. The leaf EOs and the genetic material (DNA) were extracted from samples cultivated and commercial samples. The chemical composition of the essential oils from samples of *C. verum* cultivated (Cve1-Cve5) and commercial (Cve6-c-Cv9-c) was grouped by multivariate statistical analysis of Principal Component Analysis (PCA). The major compounds were rich in benzenoids and phenylpropanoids, such as eugenol (0.7–91.0%), benzyl benzoate (0.28–76.51%), (*E*)-cinnamyl acetate (0.36–32.1%), and (*E*)-cinnamaldehyde (1.0–19.73%). DNA barcodes were developed for phylogenetic analysis using the chloroplastic regions of the *matK* and *rbcL* genes, and *psbA*-*trnH* intergenic spacer. The *psbA*-*trnH* sequences provided greater diversity of nucleotides, and *matK* confirmed the identity of *C. verum*. The combination of DNA barcode and volatile profile was found to be an important tool for the discrimination of *C. verum* varieties and to examine the authenticity of industrial sources.

## 1. Introduction

The *Cinnamomum* Schaeff genus belongs to the Lauraceae family and comprises 336 evergreen aromatic trees and shrubs distributed in Asia, Australia, and the Pacific Islands [1,2,3]. Many of these species have high economic importance and are used as an ingredient in several food products to provide flavor and aroma [4,5].

*Cinnamomum verum* J. Presl. (syn: *C. zeylanicum* Blume), also known as “true cinnamon” or “Ceylon cinnamon”, is native to Sri Lanka and southern India but also distributed in Southeast Asia, China, Burma, Indonesia, Madagascar, the Caribbean, Australia, and Africa [6]. Sri Lanka stands out for the most significant production of *C. verum* globally, corresponding to approximately 70% of the global production [7].

For a long time, *C. verum* had been widely used as a spice in traditional folk medicine and culinary practices [8]. This species has received more attention in the last decades due to increasing scientific evidence on its potential medicinal and therapeutic value [9]. Among the most relevant biological activities are anticancer [9], antidiabetic [10,11], antioxidant, anticholinergic [12], anti-inflammatory [13], anti-human immunodeficiency virus (HIV) [14], antimicrobial [15], and cytotoxic activities [16].

Such properties are attributed to unique secondary metabolic profiles of *C. verum*, including cinnamaldehyde, eugenol, cinnamyl acetate, methyl cinnamate, (*E*)-caryophyllene, and linalool [8,17,18]. Coumarin, a hepatotoxic and carcinogenic compound, is present in low concentrations in *C. verum*, and is precisely what chemically distinguishes it from other *Cinnamomum* species, such as *C. burmannii*, *C. loureiroi*, and *C. cassia* (syn. *C. aromaticum*) [13,19]. These species, known as “false cinnamon” or “cassia cinnamon” can be considered “inferior” to *C. verum* in several aspects, including their biochemical composition [5].

Systematics of *Cinnamomum* species primarily depend on morphological character analysis, which is often difficult due to its vast diversity, genetic variation, morphological similarity between species, and strict seasonality in flowering and fruiting [20]. In addition to morphology, chemotaxonomy is a supportive tool for systematics, and the volatile chemical profile, in particular, has been utilized in complex plant groups, such as Lauraceae [21,22,23].

Most medicinal plants used in the raw herbal trade are often marketed as dry twigs, powder, or billets and thus are usually difficult to identify morphologically [24]. Therefore, more scientific and accurate identification methods are required. Chromatographic fingerprinting provides an entire profile of the global components of herbal medicines and is considered to be an important method for evaluating the quality of herbal medicines [25].

There has been tremendous progress with molecular markers known as DNA barcodes in the last ten years. Currently, DNA barcodes can be combined with other technologies, such as molecular, chromatographic, and spectrum technologies, to obtain more satisfactory identification results [26]. Barcode sequences have been used to detect adulteration of cinnamon and identify high-quality species for potential cultivation purposes [27]. This study reports the integrated use of gas chromatography-mass spectrometry (GC-MS) chemotaxonomy and universal barcoding regions (*rbcL*, *matK*, and *psbA-trnH*) to assess their combined and separate identification capabilities.

## 2. Results and Discussion

### 2.1. Chemical Composition and Multivariate Analysis

The yields and volatile compositions of the *C. verum* oils of cultivated and commercial samples are displayed in Table 1.

In general, phenylpropanoids (9.25–96.14%) and benzenoid compounds (0.53–78.71%) predominated in the cinnamon oils analyzed, followed by oxygenated sesquiterpenes (0.22–14.07%), sesquiterpene hydrocarbons (1.89–8.98%), and monoterpene hydrocarbons (0.63–8.08%), with minor amounts. The main constituents were eugenol (0.7–91.0%), benzyl benzoate (0.28–76.51%), (*E*)-cinnamyl acetate (0.36–32.1%), and (*E*)-cinnamaldehyde (1.0–19.73%). Caryophyllene oxide (0–7.54%), spathulenol (0.05–3.95%), eugenyl acetate (0–3.78%), linalool (0–3.69%), and bicyclogermacrene (0–3.25%) also were identified in minor concentrations (Table 1).

The multivariate analyses of PCA (Principal Component Analysis) were applied to the constituents present in oils above 3% to evaluate the chemical variety among cultivated (Cve1-Cve5) and commercial (Cve6-c-Cv9-c) samples of *C. verum* (see Figure 1). The PCA of the constituents of oil samples explained 89.83% of the data variance and showed four main groups PC1, PC2 and PC3. PC1 accounted 42.21 % and showed a positive correlation observed for samples rich in benzyl benzoate (0.33313), (*E*)-cinnamaldehyde (0.29201), (*E*)-cinnamyl acetate (0.40693), spathulenol (0.38234), caryophyllene oxide (0.30473), linalool (0.03381), and a negative correlation with eugenol (−0.50031) and eugenol acetate (−0.38427). On the other hand, the PC2 component explained 26.43 % of the chemical variability, presenting a positive correlation with eugenol (0.1918), eugenol acetate (0.21607), (*E*)-cinnamyl acetate (0.06106), spathulenol (0.43412), and caryophyllene oxide (0.47984), and a negative correlation with linalool (−0.61597), (*E*)-cinnamaldehyde (−0.10138), and benzyl benzoate (−0.3231). Finally, the third component PC3 explained 21.19% and displayed positive correlations with the variables (*E*)-cinnamaldehyde (0.62778), eugenol (0.7923), (*E*)-cinnamyl acetate (0.48448), and eugenol acetate (0.03431) and negative with linalool (−0.20501), spathulenol (−0.22885), caryophyllene oxide (−0.31166), and benzyl benzoate (0.41495). Group I, composed of CVe2 and CVe7-c, comprised samples rich in benzyl benzoate (76.51% and 68.16%). The Cve4, Cve5, Cve6-c, and Cve9-c samples, Group II, showed a high concentration of eugenol (54.51–91.0%) followed by benzyl benzoate (0.28–22.96%). Group III, which included the Cve3 sample, was characterized by benzyl benzoate (44.14%), (*E*)-cinnamyl acetate (14.94%), caryophyllene oxide (7.54%), and spathulenol (3.95%). Cve1 and Cve8-c oils, Group IV, showed similar concentrations for (*E*)-cinnamyl acetate (32.11%, 26.15%, respectively), benzyl benzoate (15.83%, 47.68%), and (*E*)-cinnamaldehyde (19.74%, 10.82%).

Existing oil content variations can be attributed to genetic and environmental factors, including ecotype, chemotype, phenophases, and the environment [18,28]. The chemical diversity of *C. verum* has been reported in several studies, revealing the existence of four chemotypes, eugenol, eugenol and safrole, benzyl benzoate, and linalool-rich oils [18,29,30,31].

The highest oil content (1.9–2.5%) was observed for the samples (Cve4, Cve5, and Cve9-c) that presented the chemotype eugenol (>85%). The leaves of two varieties of *C. verum* chemotype-eugenol, collected in Sri Lanka, showed yields ranging from 2% to 3.90% [32,33]. The leaves of a 2-year-old *C. verum*, collected in China, with a high percentage of eugenol (>90%), presented a yield of 5.81% [34]. However, two specimens collected in the Amazon biome of chemotype eugenol (2.2%) and chemotype benzyl benzoate (2.4%) did not show significant differences between yields [18].

Therefore, one should consider that these chemotypes may result from the plant genotype, considering the season, the weather, and the collection site [35]. Eugenol is the most common chemotype identified in *C. verum* leaves [18,33,36]. In Sri Lanka, Malaysia, Cameroon, and India, the eugenol concentrations of the leaves of *C. verum* are around 80–90% [32,33,37,38,39]. On the other hand, more significant variations in eugenol content (64.20–95.0%) are observed in the specimens of *C. verum* collected in Brazil [18,30,40,41]. The samples collected in Brazil in the cities of Belém (PA), Manaus (AM), and São Luis (MA) showed predominance in eugenol (60.0–93.6%) and (*E*)-caryophyllene (1.4–8.3%). In the International Organization for Standardization (ISO) list [42], the profile of the oil of *C. verum* leaf growing mainly in Sri Lank is composed of eugenol (70–83%), benzyl benzoate (2–4.0%) and eugenyl acetate (1.3–3.0%).

(*E*)-Cinnamaldehyde and cinnamyl acetate were also significant constituents in the leaves and flowers of *C. verum* from Benin and India, respectively [43,44], and in other *Cinnamomum* species, such as *C. osmophloeum* from Taiwan [45]. In addition, these compounds are common to major constituents of *C. verum* bark oil [46,47].

Chemotypes rich in benzyl benzoate have also been reported [18,35,48]. The compositions of the leaf oil of two *C. verum* specimens from Santa Inês (MA, Brazil) revealed the existence of two chemotypes. Type I leaf oil was rich in benzyl benzoate (95.3%), linalool (1.4%), and eugenol (0.8%), whereas type II showed benzyl benzoate (65.4%), followed by linalool (5.4%), (*E*)-cinnamaldehyde (4.0%), α-pinene (3.9%), β-phellandrene (3.4%), and eugenol (3.4%) [18,35]. Benzyl benzoate (65.42%), linalool (10.81%), and (*E*)-caryophyllene (6.92%) were the major compounds of *C. verum* specimens from India [48].

The concentration of linalool in our study varied from 2.66% to 3.69%. However, the higher the concentration of this compound, the greater the flavor and fragrance, making the oil more commercially valuable [49]. The leaves of a specimen collected in Manaus (AM, Brazil) presented a linalool content of 7.0% [41]. The presence of caryophyllene oxide in the CVe3 sample collected in Belém (PA) may indicate the process of plant maturation, since (*E*)-caryophyllene may oxidize into caryophyllene oxide [29].

Cinnamon is a natural component showing a wide range of pharmacological functions; among the biological properties assigned to the majority of compounds, we can cite the chemotypes rich in eugenol and benzyl benzoate, which have antifungal and antioxidant potential [18]. *C. verum* oil, especially rich in cinnamaldehyde, can act in synergism with antibiotics commercial to increase antimicrobial potential [15]. The cinnamaldehyde compound and its derivatives act as a high anti-carcinogenic agent [9]. Finally, antidiabetic, antioxidant, and antimicrobial activities were reported for (*E*)-cinnamyl acetate [44].

**Table 1 molecules-27-07337-t001:** Volatile compositions of *Cinnamomum verum* leaf essential oils.

Oil Yield (%)			0.54	1.67	1.30	1.90	2.50	0.70	0.50	0.80	2.50
Constituent (%)	IR_(L)_	IR_(C)_	CVe1	CVe2	CVe3	CVe4	CVe5	CVe6-c	CVe7-c	CVe8-c	CVe9-c
Ethylbenzene	857 ^1^	859				0.01					
Styrene	891 ^1^	891	0.07	0.01	0.02	0.04		0.01	0.01	0.01	
Tricyclene	921 ^2^	922	0.02		0.01			0.01	0.01		
α-Thujene	924 ^2^	925	0.06	0.02	0.01			0.06	0.06	0.03	
α-Pinene	932 ^2^	933	2.89	1.18	1.52	0.23	0.12	1.69	1.51	0.33	0.23
Camphene	946 ^2^	947	1.45	0.6	1.03	0.11	0.07	0.84	0.82	0.12	0.13
6-Methylheptan-2-ol	958 ^2^	950					0.14				
Benzaldehyde	952 ^2^	957	2.36	0.51	2.13	0.11	0.15	0.45	0.8	0.44	0.26
Sabinene	969 ^2^	972	0.04	0.02	0.01		0.03	0.08	0.05	0.01	0.02
β-Pinene	974 ^2^	976	1.36	0.59	0.87	0.1	0.1	0.76	0.85	0.15	0.15
Myrcene	988 ^2^	989	0.28	0.17	0.07	0.01	0.02	0.22	0.3	0.06	0.02
Mesitylene	994 ^2^	993						0.01			
*n*-Decane	1000 ^2^	998			0.01		0.02	0.02			0.01
α-Phellandrene	1004 ^2^	1004	0.48	0.25	0.03	0.07	0.03	0.17	0.74	0.57	
δ-3-Carene	1008 ^2^	1010	0.04		0.03			0.01	0.03	0.03	
α-Terpinene	1014 ^2^	1016	0.02	0.04			0.03	0.3	0.15	0.01	0.03
*p*-Cymene	1020 ^2^	1023	0.34	0.1	0.53	0.04	0.06	0.11	0.28	0.26	0.02
β-Phellandrene	1025 ^2^	1027					0.29	1.45			0.32
Limonene	1024 ^2^	1027	0.96	0.66	0.59	0.07			1.34	0.18	
1,8-Cineole	1026 ^2^	1030				0.01					
Benzyl alcohol	1026 ^2^	1031	1.06	0.82	0.71		0.06	0.63	0.67	0.26	0.03
(*Z*)-β-Ocimene	1032 ^2^	1035	0.01	0.1				0.04	0.05		
Butyl 2-methylbutyrate	1042 ^1^	1038							0.02		
Salicylaldehyde	1039 ^2^	1040	0.03		0.03						
(*E*)-β-Ocimene	1044 ^2^	1045	0.02		0.07			0.13	0.89	0.01	
γ-Terpinene	1054 ^2^	1056	0.02					0.05	0.04	0.01	
Acetophenone	1059 ^2^	1063			0.04						
*cis*-Linalool oxide (furanoid)	1067 ^2^	1070	0.01					0.02			0.01
*trans*-Linalool oxide (furanoid)	1084 ^2^	1087									0.04
Terpinolene	1086 ^2^	1087	0.09	0.08				0.09	0.12	0.06	
Methyl benzoate	1088 ^2^	1094	0.01						0.02		
Linalool	1095 ^2^	1099	1.97	3.66	0.05		0.54	3.13	3.69	0.86	2.26
2-Methylbutyl 2-methylbutyrate	1100 ^2^	1103	0.04		0.03		0.02		0.09	0.02	0.02
α-Campholenal	1122 ^2^	1124			0.09						
(*trans*)-*p*-Menth-2-en-1ol	1136 ^2^	1137						0.01	0.01		
(*trans*)-Pinocarveol	1135 ^2^	1137			0.16						
Camphor	1141 ^2^	1143	0.03	0.03	0.04			0.01	0.04		
Pinocarvone	1160 ^2^	1161			0.36						
Hydrocinnamaldehyde	1599 ^2^	1160	1.4	0.16		0.01	0.18	0.08	0.2	0.45	
Benzyl acetate	1157 ^2^	1162	0.72	0.12			0.03	0.75	0.04	0.1	0.04
Borneol	1165 ^2^	1164	0.26	0.14	0.22		0.03	0.11	0.16	0.01	0.05
Pyruvophenone	1169 ^1^	1165			0.14						
Ethyl benzoate	1169 ^2^	1169	0.11	0.06	0.04			0.05	0.04	0.03	
Terpinen-4-ol	1174 ^2^	1176	0.1	0.09	0.09		0.05	0.1	0.14	0.04	0.04
Naphthalene	1178 ^2^	1181				0.04					
*p*-Cymen-8-ol	1179 ^2^	1183			0.02						
α-Terpineol	1186 ^2^	1189	0.34	0.32	0.3	0.03	0.05	0.18	0.43	0.15	0.11
(4*Z*)-Decenal	1193 ^2^	1192	0.08					0.01			
Myrtenol	1194 ^2^	1195	0.03		0.12						
Methyl chavicol	1195 ^2^	1197							0.01		
(Z)-Cinnamaldehyde	1217 ^2^	1217	0.17						0.03	0.06	
Hydrocinnamyl alcohol	1124 ^2^	1228	0.13		0.22			0.02			0.05
Chavicol	1247 ^2^	1253				0.03	0.11	0.05			0.06
2-Phenylethyl acetate	1254 ^2^	1255			0.05			0.03			
(*E*)-Cinnamaldehyde	1267 ^2^	1274	19.74	2.4	3.51	1	3.03	3.04	3.76	10.82	4.86
(*E*)-Cinnamyl alcohol	1303 ^2^	1303	0.25		1.27					0.04	
Isobutyl benzoate	1327 ^2^	1327	0.01		0.04				0.01		
δ-Elemene	1335 ^2^	1337	0.07					0.04	0.19	0.06	
α-Cubebene	1345 ^2^	1349	0.09		0.04				0.04		
Eugenol	1356 ^2^	1356	1.07	2.13	0.7	91	90.15	54.51	1.32	1.69	85.68
Hydrocinnamyl acetate	1366 ^2^	1370	2.91	0.29	1.3	0.03	0.13	0.14	0.26	1.26	0.1
Butyl benzoate	1376 ^1^	1371	0.05		0.06				0.04	0.02	
α-Copaene	1374 ^2^	1376	2.24	0.23	2.9	0.39	0.24	0.09	1.32	0.58	0.36
Geranyl acetate	1379 ^2^	1382	0.05								
β-Bourbonene	1387 ^2^	1384			0.07						
(*Z*)-Cinnamyl acetate	1388 ^2^	1387			0.53						
β-Cubebene	1387 ^2^	1390	0.06						0.05	0.02	
β-Elemene	1389 ^2^	1392	0.14					0.02	0.14	0.05	
Methyl eugenol	1403 ^2^	1403					0.18			0.06	0.13
*cis*-α-Bergamotene	1411 ^2^	1414			0.05						
(*E*)-Caryophyllene	1417 ^2^	1420	2.75	1.53	0.57	1.55	1.02	2.68	2.72	1.45	1.96
2-Methylbutyl benzoate	1438 ^2^	1436	0.01		0.03				0.05	0.03	
Aromadendrene	1439 ^2^	1438	0.05			0.02		0.04	0.11	0.01	0.03
(*E*)-Cinnamyl acetate	1443 ^2^	1452	32.1	4.43	14.94	0.33	0.14	0.82	1.2	26.15	0.36
α-Humulene	1452 ^2^	1454	0.7	0.32	0.09	0.28	0.21	0.52	0.68	0.33	0.38
9-*epi*-(*E*)-Caryophyllene	1464 ^2^	1460	0.05		0.03				0.04		
Cadina-1(6),4-diene	1475 ^2^	1474	0.05		0.02				0.02		
γ-Muurolene	1478 ^2^	1477	0.03		0.05		0.04	0.02	0.09	0.02	0.05
Germacrene D	1484 ^2^	1481	0.08				0.12	0.07		0.83	0.07
2-Phenylethyl 2-methylbutanoate	1486 ^2^	1485			0.11				0.02		
(*E*)-Muurola-4(14),5-diene	1493 ^2^	1492	0.05						0.03		
β-Selinene	1489 ^2^	1494			0.09						
Viridiflorene	1496 ^2^	1494					0.09				0.14
Bicyclogermacrene	1500 ^2^	1497	1.4	0.73		0.14		0.76	3.25	1.35	
α-Muurolene	1500 ^2^	1500	0.03	0.02	0.06			0.02	0.03	0.02	0.02
β-Bisabolene	1505 ^2^	1508			0.04						
γ-Cadinene	1513 ^2^	1514		0.08	0.15		0.04	0.02	0.02	0.06	0.07
Cubebol	1514 ^2^	1515	0.04								
δ-Cadinene	1522 ^2^	1523	0.33	0.29	0.53	0.04	0.13	0.1	0.25	0.19	0.23
Eugenol acetate	1521 ^2^	1526				3.78	0.26	1.48			0.19
(*E*)-*o*-Methoxycinnamaldehyde	1527 ^2^	1527	0.03							0.03	
(*E*)-Cadina-1,4-diene	1533 ^2^	1532	0.03								
α-Cadinene	1537 ^2^	1537			0.02						
α-Calacorene	1544 ^2^	1542			0.03						
Caryolan-8-ol	1571 ^2^	1569							0.02		
Spathulenol	1577 ^2^	1577	0.81	0.15	3.95	0.05	0.06	0.19	0.64	0.8	0.07
Caryophyllene oxide	1582 ^2^	1582	0.61	0.31	7.54	0.17	0.56	0.41	0.67	0.52	0.53
β-Copaen-4α-ol	1590 ^2^	1586			0.34						
Viridiflorol	1592 ^2^	1591	0.03					0.02	0.13	0.03	
Cubeban-11-ol	1595 ^2^	1593							0.03		
Rosifoliol	1600 ^2^	1601							0.08		
Humulene epoxide II	1608 ^2^	1608	0.1		0.83		0.06	0.03	0.06	0.06	0.05
1-*epi*-Cubenol	1627 ^2^	1627	0.06		0.17				0.04	0.02	
Caryophylla-4(12),8(13)-dien-5-α-ol	1639 ^2^	1631			0.05						
Caryophylla-4(12),8(13)-dien-5-β-ol	1639 ^2^	1635	0.02		0.43				0.02	0.08	
α-Muurolol (=Torreyol)	1644 ^2^	1640		0.14							
*epi*-α-Murrolol (=τ-muurolol)	1640 ^2^	1640									0.08
*epi*-α-Cadinol (=τ-cadinol)	1638 ^2^	1640			0.2			0.04		0.06	
Cubenol	1645 ^2^	1642	0.06						0.07		
α-Cadinol	1652 ^2^	1654	0.03	0.12			0.03	0.04	0.03	0.07	0.07
14-Hydroxy-9-*epi*-(*E*)-caryophyllene	1668 ^2^	1670			0.45						
Mustakone	1676 ^2^	1676			0.07						
Khusinol	1675 ^2^	1684			0.12						
Amorpha-4,9-dien-2-ol	1704 ^2^	1698			0.04						
Benzyl benzoate	1759 ^2^	1769	15.83	76.51	44.11	0.29	1.22	22.96	68.16	47.68	0.28
Phytone	1841 ^1^	1843							0.02	0.02	
2-Phenylethyl benzoate	1856 ^1^	1852		0.37	2.43			0.03	0.23	0.11	
Benzyl salicylate	1864 ^2^	1866		0.15	0.03				0.02		
Phytol	2110 ^1^	2110								0.06	
Monoterpene Hydrocarbons			8.08	3.81	4.77	0.63	0.75	6.01	7.24	1.83	0.92
Oxygenated Monoterpenes			2.79	4.24	1.45	0.04	0.67	3.56	4.47	1.06	2.51
Sesquiterpene Hydrocarbons			8.15	3,2	4.74	2.42	1.89	4.38	8.98	4.97	3.31
Oxygenated Sesquiterpenes			1.76	0.72	14.07	0.22	0.71	0.73	1.79	1.64	0.8
Phenylpropanoids			56.4	9.25	21.94	96.14	93.89	60.01	6.57	40.11	91.37
Benzenoids			21.66	78.71	50.46	0.53	1.75	25.05	70.32	49.13	0.67
Others			0.12		0.2		0.18	0.03	0.13	0.1	0.03
Total Identifield			98.96	99.93	97.63	99.98	99.84	99.77	99.5	98.84	99.61

^1^ [50]; ^2^ [51]; RI_(C)_: Calculated retention index; RI_(L)_: Literature retention index; *C. verum* cultivated (Cve1–Cve5); *C. verum* commercials (Cve6-c–Cve9-c).

### 2.2. DNA Authentication and Genetic Variability of Cinnamomum verum Samples

A DNA barcode was incorporated to address the challenge of chemical plasticity, as the genetic makeup of a particular species should be more stable under various environmental conditions [52]. In plants, the establishment and refinement of DNA barcodes have been more challenging due to the distinct genetic diversity among different species [53].

PCR amplification and sequencing success are important factors for selecting ideal barcode loci [54,55]. The sequences of the *matK*, *rbcL*, and *psbA-trnH* regions were constructed for all studied samples, including five specimens of *C. verum* (Cve1, Cve2, Cve3, Cve4, Cve5) which had confirmed morphological identity, and four commercial samples (Cve6-c, Cve7-c, Cve8-c, Cve9-c). The PCR success rate was 100% for all the analyzed loci except *matK*, which did not show any amplification in two commercial samples (CVe7-c and CVe9-c). All PCR products were successfully sequenced, and high-quality bidirectional sequences were obtained.

The success of species identification depends on the quality of the barcode sequence and the taxonomic coverage of reference sequences in the GenBank database [56]. The sequence with the highest homology, maximum query coverage, and maximum score was used as a reference to assign the identity of the species. The herbal analysis of the market samples revealed that all of *C. verum* samples were authentic. Using authenticated raw materials is the basic starting point in developing safe and high-quality natural health products. The possibility of adulteration is high due to misidentification because the collectors do not have the taxonomic expertise to differentiate morphologically similar species [57].

In relation to the regions, homology searches using the NCBI Blast program found matched sequences with *C. verum*, resulting in species-level identification for the *matK* region. On the other hand, the *psbA-trnH* and *rbcL* sequences provided identification only at the genus level. Another important criterion of an ideal barcode is its discriminatory power [55,58].

The *psbA-trnH* sequences had the highest nucleotide diversity (π: 0.01449), polymorphic sites (20 bp), and parsimonious-informative characters (6 bp). The intergenic spacer is described as a DNA barcode rich in simple sequence repeats and small insertions and deletions (INDELs) [59,60]. In contrast, the sequences of *matK* and *rbcL* coding regions showed a phylogenetically conserved nature, with low nucleotide diversity (0.00000–0.000123), polymorphic sites (0 and 3 bp), and parsimonious-informative (0 bp) (Table 2). The alignment of the concatenated matrix (*rbcL+matK+psbA-trnH*) presented a total of 1699 bp of characters, of which 6 bp are considered informative, and 23 bp are polymorphic sites (Table 2).

A multi-locus approach of barcode regions was used to establish the DNA barcode signatures from commercialized and cultivated of *C. verum* samples in the Amazon. Interestingly, cultivated samples in different locations (Benevides, Belém, Curuçá and Maranhãozinho) showed great genetic similarities with commercial samples. The alignment of the concatenated matrix showed great genetic variability in the *psbA*-*trnH* intergenic region compared to *matK* and *rbcL* in samples obtained (Figure 2).

Due to the greater nucleotide diversity, we use the *pbsA*-*trnH* region to check the distances between sequences. The genetic distances were low, with a mean of 0.015 (Appendix A), indicating little genetic variability between specimens of *C. verum*. DNA barcodes have also been suggested to discriminate species and identify adulterants in *Cinnamomum*. For example, commercial samples of cinnamon were identified as adulterants in *C. aromaticum* and *C. malabathrum* using sequences of *rbcL, matK*, and *psbA-trnH* [27]. Nevertheless, these same regions used individually or in combination did not show sufficient genetic variation to discriminate *C. capparu-coronde*, *C. citriodorum*, *C. litseifolium*, *C. sinharajaense*, *C. ovalifolium*, and *C. verum* species in Sri Lanka [61].

### 2.3. Molecular and Chemical Methods

Most *Cinnamomum* plants are highly economically valuable tree species. However, *Cinnamomum* species share similar morphological features in their taxonomy. Thus, developing a rapid and feasible method for the identification of *Cinnamomum* plants is needed to prevent their adulteration of trees [62]. DNA barcodes can be incorporated to address the challenge of chemical plasticity, as the genetic makeup of a particular species should be more stable under various environmental conditions [52].

The DNA barcode can only authenticate the medicinal plant while the chemical profile provides information on the presence and concentration of compounds with pharmacological activity [63]. This diversity of compounds is generally determined by the genetic constitution of the plant, although environmental factors may also influence the type, amount, and concentrations of the compounds present in the essential oil [64].

Chemical compounds commonly occur similarly in members of the same phylogenetic clade, and their presence or absence may indicate the common origin and, therefore, lineage [65]. The differences/fluctuations in the composition of secondary metabolites could be due to genetic modifications linked to the adaptation of these plant species to their environment [66]. Our phylogenetic analysis study allowed specific taxonomic identifications up to the level of *C. verum* varieties.

The complementary use of chemical and molecular markers for quality control achievement of the *C. verum* species and other plant materials should be tested in commercialized leaves and in herbal preparations. The identification of adulterants, fillers and/or substitutes could be accomplished only if the molecular databases of medicinal plants are enriched with more studies [67,68].

## 3. Materials and Methods

### 3.1. Plant Material

Leaf samples of five cultivated *Cinnamomum verum* species were collected in Belém (PA, Brazil), Benevides (PA, Brazil), Curuçá (PA, Brazil), and Maranhãozinho (AM, Brazil). The plant vouchers were identified and cataloged in the Herbarium João Murça Pires, Emilio Goeldi Museum, Pará state, Brazil, as listed in Table 1. Four commercial samples of cinnamon leaves were purchased in local markets of companies in Belém (PA, Brazil) and labeled as Cve-6c to Cve9-c to check the authenticity of the commercialized product (Table 3).

### 3.2. Essential Oil Extraction

The leaves were dried for two days at room temperature and then subjected to essential oil distillation. The dry leaves were pulverized and submitted to hydrodistillation using a Clevenger-type apparatus (3 h). The oils were dried over anhydrous sodium sulfate, and the yields were calculated based on the dry weight of the plant material. The moisture content of each sample was measured using an infrared moisture balance ID50 with a heat source (Marte^®^, Santa Rita do Sapucaí, MG, Brazil). The moisture content of each sample was measured using an infrared moisture balance. The procedure was performed in triplicate.

### 3.3. GC-MS and GC(FID) Analysis

The oil samples were analyzed on a GCMS-QP2010 Ultra system (Shimadzu Corporation, Tokyo, Japan), equipped with an auto-injector (AOC-20i). The parameters of analysis were: A silica capillary column Rxi-5ms (30 m × 0.25 mm; 0.25 μm film thickness) (Restek Corporation, Bellefonte, PA, USA); injector temperature: 250 °C; oven temperature programming: 60–240 °C (3 °C/min); helium as carrier gas, adjusted to a linear velocity of 36.5 cm/s (1.0 mL/min); splitless mode injection of 1 μL of the sample (oil 5 μL:hexane 500 μL); ionization by electronic impact at 70 eV; ionization source and transfer line temperatures at 200 °C and 250 °C, respectively. The mass spectra were obtained by automatically scanning every 0.3 s, with mass fragments in the range of 35–400 m/z. The quantitative data regarding the volatile constituents were obtained by peak-area normalization using a GC 6890 Plus Series (Agilent, Wilmington, DE, USA), coupled to a flame ionization detector (FID), operated under similar GC-MS system conditions. The retention index was calculated for all volatile components using a homologous series of C_8_-C_20_ *n*-alkanes (Sigma-Aldrich, St. Louis, MO, USA), according to the linear equation of Van den Dool and Kratz [69]. The components of oils were identified by comparing their retention indices and mass spectra (molecular mass and fragmentation pattern) with data stored in the [28,29,70] libraries.

### 3.4. Multivariate Statistical Analysis of Chemical Composition

The chemical compositions of the leaf samples with a percentage above 3% were used as variables in multivariate analysis. First, the matrix’s data standardization was performed by subtracting the mean and dividing it by the standard deviation. The Principal Component Analysis was applied to verify the interrelation in the oil’s components (OriginPro trial version, OriginLab Corporation, Northampton, MA, USA).

### 3.5. DNA Isolation, PCR Amplification, and Sequencing

Genomic DNA material was extracted from 100 mg of dried leaf tissue of each plant using a plant DNA isolation Kit (PureLink™ Genomic DNA, Invitrogen, Carlsbad, CA, USA) according to the protocol given by the company and stored at −20 °C. Three chloroplast DNA regions were used for amplification: *rbcL*, *matK*, and the intergenic spacer *psbA*-*trnH*. The Consortium for the Barcode of Life’s (CBOL) plant working group recommended using a core of a two-locus combination of *rbcL* + *matK* as the plant barcode, with *psb*A-*trnH* as complementary sequences [55]. Polymerase chain reactions (PCR) were performed in a volume of 50 µL containing 35.0 μL of ultrapure water (Invitrogen, Carlsbad, CA, USA), 5 μL of 10x Advantage Buffer (200 mM Tris-HCl pH 8.4; 500 mM KCl, Invitrogen, Carlsbad, CA, USA), 1 μL of deoxynucleotide (dNTP) (10mM, Biotium, Fremont, CA, USA), 0.5 μL of Taq DNA polymerase (5 U/μL, Invitrogen, Carlsbad, CA, USA), 4.0 μL template DNA (at a concentration of approximately 20 ng/μL) and 1 µL of each primer, forward and reverse (10 mM), synthesized by the companies Síntese Biotecnologia (Belo Horizonte, Brazil). DNA amplifications were conducted in a thermocycler (GeneAmp PCR System 9700, Foster, CA, USA), and the negative control was carried out for all PCR reactions in the absence of DNA. Amplification products were visualized in agarose gel 1.5% and, subsequently, the amplicons were sent for purification, quantification and sequencing at the company ACTgene Análises Moleculares Ltd. (Alvorada, Brazil). Table 4 presents the sequences of the primers of each fragment and its PCR amplification conditions.

### 3.6. Sequence Identity and Distance Genetics Analysis

The forward and reverse sequences of each amplified region (*matK*, *rbcL*, and *psbA-trnH*) were edited and aligned using the software MUSCLE algorithm [75] implemented within MEGA 7 software [76]. Sequences were compared with available sequences in the National Center for Biotechnology Information (NCBI) GenBank database (http://www.ncbi.nlm.nih.gov/, accessed in 1 June 2022), using the tool Blast N. DNA sequences generated in this study were deposited in the NCBI GenBank, and accession numbers are listed in the Supporting Information (Table 5).

The sequences of *rbcL*, *matK*, and *psbA*-*trnH* were analyzed in DnaSP v6 [77] to obtain the median length described the genetic variability of each marker (bp) and total alignment length (bp), both discounting gaps, the number of sites with gaps, and nucleotide diversity (π). The sequencer was concatenated using the program Phylosuite [78] and aligned with the CLUSTAL W [79] in Mega software. The alignment was edited in BIOEDIT program [80]. The *pbsA*-*trnH* region was used to estimate the pairwise distance using the Kimura two-parameter (K2P) model [81].

## 4. Conclusions

We developed a pioneering study by integrating the volatile profile and molecular sequences for rapid authentication and discrimination of *C. verum* samples in this study. The essential oils of the samples with occurrence in the Amazon were rich in benzenoids and phenylpropanoids. The wide array of volatile chemical structures identified in the samples and their distribution pattern was utilized to differentiate chemotypes, such as (E)-cinnamyl acetate, benzyl benzoate, (E)-cinnamaldehyde, caryophyllene oxide, spathulenol, linalool, and eugenol. The species identity has been confirmed using barcode sequences, which is crucial for commercial samples with morphological data limitations.

## Figures and Tables

**Figure 1 molecules-27-07337-f001:**
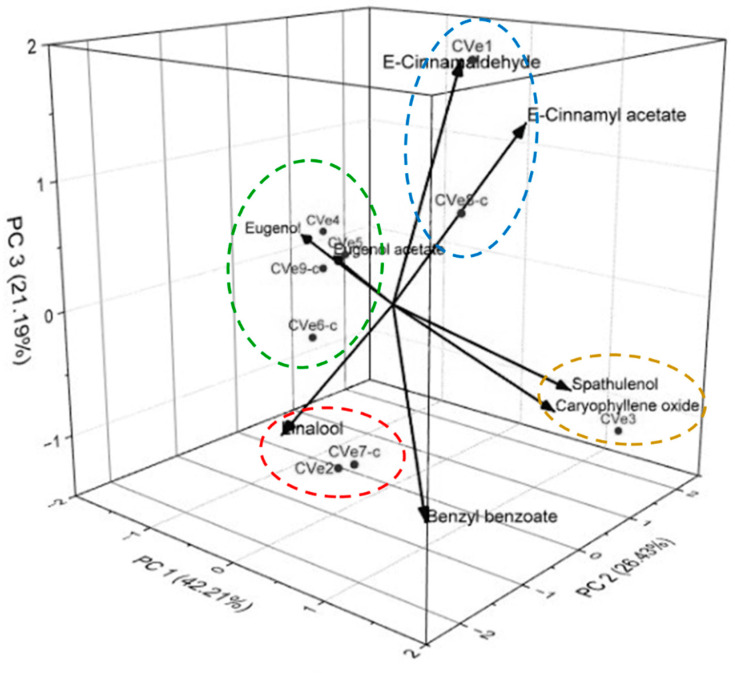
The tridimensional plot of the first three components (PC1, PC2 and PC3) from principal component analysis (PCA) of *Cinnamomum verum* samples, based on the main constituents present in the analyzed essential oils: cultivated (Cve1-Cve5) and commercial samples (Cv6-c-Cve9-c).

**Figure 2 molecules-27-07337-f002:**
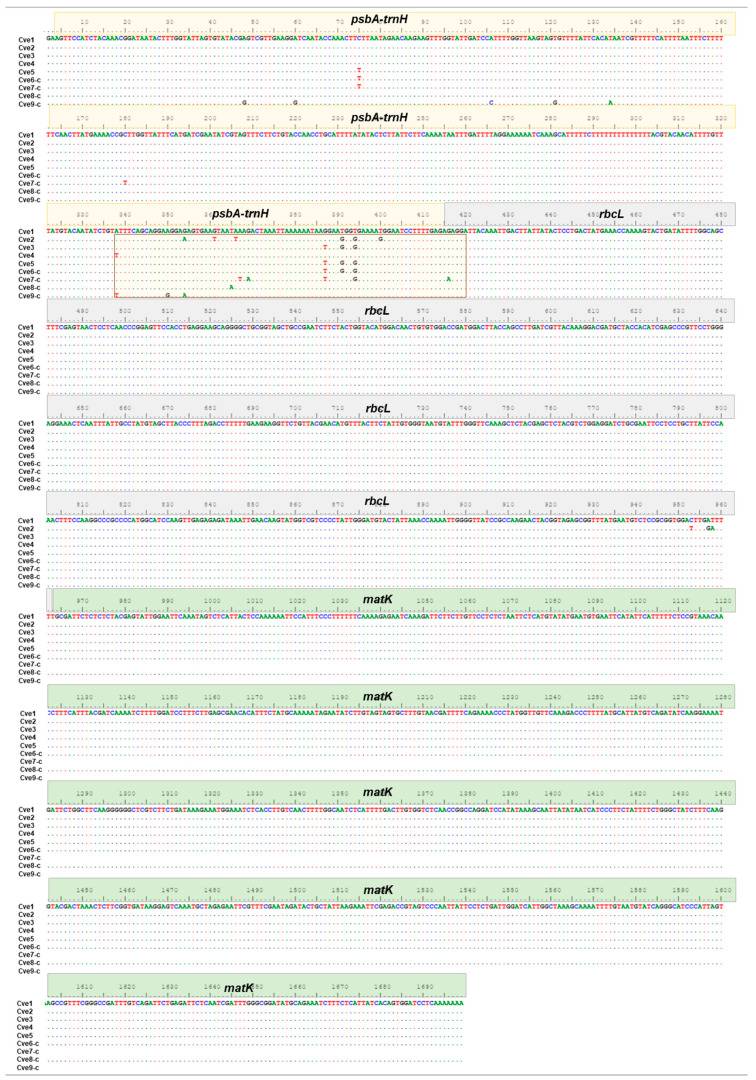
Multiple alignments of the nucleotide sequences of the markers *matK*, *rbcL* and *psbA*-*trnH* of cultivated (Cve1–Cve5) and commercial samples (Cve6-c–Cve9-c) of *Cinnamomum verum*. The *dots* indicate the consensus nucleotides, and the *Boxes* shown the variable region observed in *pbsA*-*trnH**.psbA*-*trnH* (1–431 bp), *rbcL* (432–974 bp) and *matK* (975–1712 bp).

**Table 2 molecules-27-07337-t002:** Molecular characteristics of markers evaluated for *Cinnamomum verum*.

DNA Markers	Aligned Length (bp)	Nucleotide Diversity (π)	Polymorphic Sites	Parsimony-Informative Sites
*rbcL*	543	0.000123	3	0
*matK*	738	0.00000	0	0
*psbA*-*trnH*	418	0.01449	20	6
*rbcL+matK+psbA-trnH*	1699	0.00700	23	6

**Table 3 molecules-27-07337-t003:** Data from cultivated and commercial *Cinnamomum verum* samples.

Sample Code	Type	Collection Site/Company	Voucher
Cve1	Cultivated	Maranhãozinho (MA)	243613
Cve2	Cultivated	Curuçá (PA)	243614
Cve3	Cultivated	Belém (PA)	243615
Cve4	Cultivated	Benevides (PA)	243616
Cve5	Cultivated	Belém (PA)	243617
Cve6-c	commercialized	Ver-o-peso market	Not cataloged
Cve7-c	commercialized	Tempero e Cia	Not cataloged
CVe8-c	commercialized	Ver-o-peso market	Not cataloged
Cve9-c	commercialized	Pau de verônica	Not cataloged

**Table 4 molecules-27-07337-t004:** Primer sequences applied in DNA amplification of *Cinnamomum verum* species and its experimental conditions.

Region	Primers	Sequence (5′–3′)	Amplification Protocol
*matK* ^1^	*matk 2.1*	CCTATCCATCTGGAAATCTTAG	95 °C 7min; 95 °C 1min, 53 °C 1 min, 72 °C 1 min, 35 cycles; 72 °C 7 min
	*matk 5*	GTTCTAGCACAAGAAAGTCG	
*psbA* ^2^	*psbA3_f F*	GTTATGCATGAACGTAATGCT	95 °C 7 min; 95 °C 1min, 56 °C 1 min, 72 °C 1 min, 35 cycles; 72 °C 7 min
*trnH* ^3^	*trnHf_05 R*	CGCGCATGGTGGATTCACAATCC	
*rbcL* ^4^	*rbcL*1	ATGTCACCACAAACAGAGACTAAAGC	95 °C 7min; 95 °C 1min, 53 °C 1 min, 72 °C 1 min, 35 cycles; 72 °C 7 min
	*rbcL*a	GTAAAATCAAGTCCACCRCG	

^1^ [71]; ^2^ [72]; ^3^ [73]; ^4^ [74].

**Table 5 molecules-27-07337-t005:** GenBank accession numbers of *Cinnamomum verum* species collected in the Amazon.

Sample Code	*matK*	*psbA*-*trnH*	*rbcL*
Cve1	OM981169	OM981164	OM981159
Cve2	OM981170	OM981165	OM981160
Cve3	OM981171	OM981166	OM981161
Cve4	OM981172	OM981167	OM981162
Cve5	OM981173	OM981168	OM981163

## Data Availability

Publicly available gene sequence datasets were analyzed in this study. These data can be found here: https://www.ncbi.nlm.nih.gov/genbank/; accession numbers: *C. verum* (OM981169, OM981164, OM981159, OM981170, OM981165, OM981160, OM981171, OM981166, OM981161, OM981172, OM981167, OM981162, OM981173, OM981168, OM981163).

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
