# Peer review of "Essential Oil Chemotypes and Genetic Variability of Cinnamomum verum Leaf Samples Commercialized and Cultivated in the Amazon"

_molecules, 2022, doi:10.3390/molecules27217337_

Round 1

Reviewer 1 Report

The manuscript of Júlia Karla A. M. Xavier et al entitled "Essential oil chemotypes and genetic variability of Cinnamomum verum leaf samples commercialized and cultivated in the Amazon" is valuable comparison of chemical profiles of C. verum cultivated and commercial samples. This work shows differences between samples of different origin and tries to relate them to genetic variability. However, while I have no major objections about the chemical analyzes and the methodology of estimating genetic diversity, I have doubts about the method of phylogenetic analyzes and the manner in which chemical differences have been presented in a phylogenetic context. As the authors themselves noted, obtained phylogenetic trees are just "moderately supported" (line 245). In my opinion this statement should be even more critical as Posterior probabilities lower than 0.99 make any phylogenetic estimation completely unreliable - basing on such result, any attempt to relate chemical profiles to phylogeny is objectionable.

Moreover, I have great doubts about the correctness of phylogenetic analyzes. The correct strategy in analysis of concatenated markers is: preparing independent alignments of each marker, trimming of alignment if needed, then concatenation of independent alignments. If such strategy is applied, the statistical numbers of each alignment should be additive, i.e., the length of concatenated alignment should be the sum of lengths of single marker alignments. But the numbers in the Table 2 are not additive: (1) the length of concatenated alignment should be 600+829+461=1890 (but in the Table 2 is 1911), (2) polymorphic sites 2+1+31=34 (in the Table 2 is 39), PiS 0+0+7=7 (in the Table 2 is 9). It suggests that methodology of concatenated alignment preparation was incorrect - maybe sequences of markers were concatenated first, then aligned? If the answer is yes, it was wrong strategy. Used methodology should be clarified in the text.

I recommend: (1) not to relate the chemical profiles to phylogeny, in my opinion information about the genetic variability among samples should be sufficient (or try to relate profiles to haplotypes?); (2) perform phylogenetic analyses once again according to correct scheme and decide if they should be included depending on results. I think that rooted phylogeny is not necessary - unrooted phylogeny without outgroup should be enough for reasoning (and presented as network) if obtained trees are well supported.

Minor comments:

221: In my opinion a pairwise comparison of sequence differences would be more appropriate than a tree with moderate supported clades.
234: Consensus tree in ML analysis is rather unusual - if identical sequences were obtained for samples, haplotype analysis would be even more appropriate than phylogenetic one.
244: BI instead of IB
254: samples Cve3, Cve2 and Cve7-c cannot be treated as a group because the do not form clade.
351: dNTP instead of DNTP
353: volume of DNA is not informative as the concentration of DNA in not provided; use estimated amount of DNA instead
382: I do not understand the sentence: "The nucleotide substitution model selected was partitioned by markers", I guess that it should be described like this: "The nucleotide substitution model was selected independently for each marker" if it is what authors meant. It should be also clarified if independent partitions for each marker were applied for concatenated alignment in ML an BI analysis (partitions should be used). It should also be added if independent model and model's parameters were applied for each partition.
386: The sentence "and a sampling frequency set to every 10,000,000 generations" should be changed to "and lasted for 10,000,000 generations", sampling is described in the following sentence.

Author Response

Revisor 1

The manuscript of Júlia Karla A. M. Xavier et al entitled "Essential oil chemotypes and genetic variability of Cinnamomum verum leaf samples commercialized and cultivated in the Amazon" is valuable comparison of chemical profiles of C. verum cultivated and commercial samples. This work shows differences between samples of different origin and tries to relate them to genetic variability. However, while I have no major objections about the chemical analyzes and the methodology of estimating genetic diversity, I have doubts about the method of phylogenetic analyzes and the manner in which chemical differences have been presented in a phylogenetic context. As the authors themselves noted, obtained phylogenetic trees are just "moderately supported" (line 245). In my opinion this statement should be even more critical as Posterior probabilities lower than 0.99 make any phylogenetic estimation completely unreliable - basing on such result, any attempt to relate chemical profiles to phylogeny is objectionable. 

Moreover, I have great doubts about the correctness of phylogenetic analyzes. The correct strategy in analysis of concatenated markers is: preparing independent alignments of each marker, trimming of alignment if needed, then concatenation of independent alignments. If such strategy is applied, the statistical numbers of each alignment should be additive, i.e., the length of concatenated alignment should be the sum of lengths of single marker alignments. But the numbers in the Table 2 are not additive: (1) the length of concatenated alignment should be 600+829+461=1890 (but in the Table 2 is 1911), (2) polymorphic sites 2+1+31=34 (in the Table 2 is 39), PiS 0+0+7=7 (in the Table 2 is 9). It suggests that methodology of concatenated alignment preparation was incorrect - maybe sequences of markers were concatenated first, then aligned? If the answer is yes, it was wrong strategy. Used methodology should be clarified in the text. 

Author’s: We made revisions in the analysis of nucleotic diversity and corrected the identified errors.

I recommend: (1) not to relate the chemical profiles to phylogeny, in my opinion information about the genetic variability among samples should be sufficient (or try to relate profiles to haplotypes?); (2) perform phylogenetic analyses once again according to correct scheme and decide if they should be included depending on results. I think that rooted phylogeny is not necessary - unrooted phylogeny without outgroup should be enough for reasoning (and presented as network) if obtained trees are well supported.

Author’s: We attended your suggestion and find it pertinent not to add the phylogenetic analysis.

Minor comments:

221: In my opinion a pairwise comparison of sequence differences would be more appropriate than a tree with moderate supported clades.

Author’s: We accept your suggestion and added Figure 2 representing the alignment of the sequences.

234: Consensus tree in ML analysis is rather unusual - if identical sequences were obtained for samples, haplotype analysis would be even more appropriate than phylogenetic one.

Author’s: We performed haplotype analysis and found it appropriate not to include it. The number of individuals in the sample was small to infer something in this analysis.

244: BI instead of IB

Author’s: This sentence was removed.

254: samples Cve3, Cve2 and Cve7-c cannot be treated as a group because the do not form clade.

Author’s: We accept your suggestion and remove this paragraph that related the chemical and phylogenetic profiles.

351: dNTP instead of DNTP

Author’s: This sentence was corrected.

353: volume of DNA is not informative as the concentration of DNA in not provided; use estimated amount of DNA instead

Author’s: This information was inserted.

382: I do not understand the sentence: "The nucleotide substitution model selected was partitioned by markers", I guess that it should be described like this: "The nucleotide substitution model was selected independently for each marker" if it is what authors meant. It should be also clarified if independent partitions for each marker were applied for concatenated alignment in ML a BI analysis (partitions should be used). It should also be added if independent model and model's parameters were applied for each partition.

Author’s: We remove the phylogenetic analysis.

386: The sentence "and a sampling frequency set to every 10,000,000 generations" should be changed to "and lasted for 10,000,000 generations", sampling is described in the following sentence.

Author’s: We remove the phylogenetic analysis.

Reviewer 2 Report

The authors carried out a study on the chemical composition of Cinnamomum verum essential oil entitled

"Essential oil chemotypes and genetic variability of Cinnamomum verum leaf samples commercialized and cultivated in the Amazon"

The manuscript looks interesting and contains information that may be relevant, however, the authors need major revisions in the writing and interpretation of the results.

1) the abstract only shows that the plant needs some compounds, always giving emphasis to the statistical analysis, I think the authors should put numbers showing which are the major compounds as well as which classes they belong to.

From line 47 to line 52, the authors report several biological activities that the species studied has potential for application, this leads the reader to believe that some type of study of the biological potential will be carried out, in my view, it is a mistake and should be reviewed. This is shown in other parts of the introduction. Probably the authors will report in their answer that they added these snippets to show potential applications.

Section 2.1. Chemical composition and Multivariate analysis

line 83-88 the paragraph only describes the method, if the authors have a materials and method section why perform the description in results and discussion? this should be deleted, the journal's guide for authors is clear about this.

The discussion is without depth, 1 of this MS the authors rehearse a discussion about the potential applications, even though I don't agree that it is part of the introduction I believe it can be useful to bring this information to the results and discussions section, for example, in no moment in the discussion is reported the potential application of the major compounds.

for me, the execution of the statistical analysis is not satisfactory, as the authors explain the formation of a group between samples that has 80% difference in chemical composition and only 20% of similarity. Static analysis is used to complement a study, but performing authors must analyze the data very carefully so as not to misinterpret it.

HCA is a repeated calculation of distance measures between objects. PCA defines a new orthogonal coordinate system that describes the variance in a dataset.

How is HCA linked to PCA? you use HCA to classify extract and report euclidean distances with PCA loading. I think that euclidean distances could be related to the PCA score. Otherwise, the relation you discuss between PCA loading and euclidean distances might be casual.

Dividing the original dataset as created means, in my opinion, decreasing the population of the data set and consequently the significance of the classification. While holding the entire dataset would allow applying of supervised techniques such as linear discriminant analysis (LDA) to find differences among defined classes, such as type of extracts, geographical origin, and so on, in order to find the relation with the chemical composition of plants extracts.

In my view, authors should review the input methods of statistical analysis.

Materials and methods

The samples

Nve6-c

Nve7-c

CVe8-c

Nve9-c

They were not identified, as there is no Voucher number assigned to them, if a DNA study had not been carried out for me, it could be another plant.

The conclusion needs to be improved, the word "success" is not scientific. Also, what is a chemical alliance? Could the authors explain this concept?

Author Response

Revisor 2

The authors carried out a study on the chemical composition of Cinnamomum verum essential oil entitled

"Essential oil chemotypes and genetic variability of Cinnamomum verum leaf samples commercialized and cultivated in the Amazon"

The manuscript looks interesting and contains information that may be relevant, however, the authors need major revisions in the writing and interpretation of the results.

1) the abstract only shows that the plant needs some compounds, always giving emphasis to the statistical analysis, I think the authors should put numbers showing which are the major compounds as well as which classes they belong to.

Author’s: We attended these suggestions and add information about the classes and concentrations present in the compounds.

From line 47 to line 52, the authors report several biological activities that the species studied has potential for application, this leads the reader to believe that some type of study of the biological potential will be carried out, in my view, it is a mistake and should be reviewed. This is shown in other parts of the introduction. Probably the authors will report in their answer that they added these snippets to show potential applications.

Section 2.1. Chemical composition and Multivariate analysis

line 83-88 the paragraph only describes the method, if the authors have a materials and method section why perform the description in results and discussion? this should be deleted, the journal's guide for authors is clear about this.

Author’s: we accept your suggestion and remove this paragraph.

The discussion is without depth, 1 of this MS the authors rehearse a discussion about the potential applications, even though I don't agree that it is part of the introduction I believe it can be useful to bring this information to the results and discussions section, for example, in no moment in the discussion is reported the potential application of the major compounds.

Author’s: We add a paragraph reporting the properties of the main compounds of C. verum.

for me, the execution of the statistical analysis is not satisfactory, as the authors explain the formation of a group between samples that has 80% difference in chemical composition and only 20% of similarity. Static analysis is used to complement a study, but performing authors must analyze the data very carefully so as not to misinterpret it.

HCA is a repeated calculation of distance measures between objects. PCA defines a new orthogonal coordinate system that describes the variance in a dataset.

How is HCA linked to PCA? you use HCA to classify extract and report euclidean distances with PCA loading. I think that euclidean distances could be related to the PCA score. Otherwise, the relation you discuss between PCA loading and euclidean distances might be casual.

Dividing the original dataset as created means, in my opinion, decreasing the population of the data set and consequently the significance of the classification. While holding the entire dataset would allow applying of supervised techniques such as linear discriminant analysis (LDA) to find differences among defined classes, such as type of extracts, geographical origin, and so on, in order to find the relation with the chemical composition of plants extracts.

In my view, authors should review the input methods of statistical analysis.

Author’s: We agree with this suggestion and repeat the multivariate statistical analysis applying PCA analysis to explain better the groups formed.

Materials and methods

The samples

Nve6-c

Nve7-c

CVe8-c

Nve9-c

They were not identified, as there is no Voucher number assigned to them, if a DNA study had not been carried out for me, it could be another plant. The conclusion needs to be improved; the word "success" is not scientific. Also, what is a chemical alliance? Could the authors explain this concept?

Author’s: This sentence has been corrected, and we decided to exclude "chemical alliances" since this study has no way of comparing phylogenetic and chemical profiles.

Round 2

Reviewer 1 Report

The authors took into account my previous comments, but the addition of a new figure (Figure 2) made me have serious doubts about the quality of obtained and presented sequences:

- There is a gap (312-317) in all sequences which influence the length of psbA-trnH marker as well the length of concatenated sequences. The gap should be removed and the length should be re-calculated carefully.
- Markers are mislabeled: according to the Table 2 rbcL alignment is 599 bp, but on the picture rbcL is labelled as 700 bp long; matK is 839 bp according to the Table 2 but on the picture is 738 bp long
- rbcL and matK are protein coding genes so each indel induce frame shift - it is impossible in such conserved genes. There are also indels and many substitutions at both ends of psbA-trnH marker, but one of ends is the end of coding sequence of psbA gene and it should be conserved fragment too (maybe I am wrong and coding sequence was removed?). Authors should carefully inspect obtained chromatogram files (not text files) once again, remove primers sequences (the end of matK marker looks like the sequence of matK_5 primer), correct possible errors basing on comparison to available reference sequences of Cinnamomum verum, mark questionable nucleotides as "N" (or other) if needed or trunk the sequence, check the correctness by translating nucleotide sequences to protein ones. Then re-calculation of all numbers should be done.

Minor comments:

215: bp instead of pb
219: bp instead of pb
221-222 and Table 2: 2+1+37=40 (not 39), 0+1+9=10 (not 9), but these number should be re-calculated basing on new alignments
242: psbA-trnH should be in italics
244: "indicating little genetic variability intraspecific" this part of sentence should be rewritten
363: marker name should be in italics

Author Response

The authors took into account my previous comments, but the addition of a new figure (Figure 2) made me have serious doubts about the quality of obtained and presented sequences:

- There is a gap (312-317) in all sequences which influence the length of psbA-trnH marker as well the length of concatenated sequences. The gap should be removed, and the length should be re-calculated carefully.

Author’s: We accept this suggestion. The gap was removed from the sequences, and a new alignment was redone.

- Markers are mislabeled: according to the Table 2 rbcL alignment is 599 bp, but on the picture rbcL is labelled as 700 bp long; matK is 839 bp according to the Table 2 but on the picture is 738 bp long

Author’s: A new alignment from the sequences was carried out, and the labels were corrected.

- rbcL and matK are protein coding genes so each indel induce frame shift - it is impossible in such conserved genes. There are also indels and many substitutions at both ends of psbA-trnH marker, but one of ends is the end of coding sequence of psbA gene and it should be conserved fragment too (maybe I am wrong and coding sequence was removed?). Authors should carefully inspect obtained chromatogram files (not text files) once again, remove primers sequences (the end of matK marker looks like the sequence of matK_5 primer), correct possible errors basing on comparison to available reference sequences of Cinnamomum verum, mark questionable nucleotides as "N" (or other) if needed or trunk the sequence, check the correctness by translating nucleotide sequences to protein ones. Then re-calculation of all numbers should be done.

Author’s: We attended to this suggestion and carefully inspected all markers' chromatograms, referring to the alignment and subsequent analyses. Also, we removed the psbA-trnH marker coding fragment and the matK_5 primer sequences.

Minor comments:

215: bp instead of pb

Author’s: This sentence was corrected

219: bp instead of pb

Author’s: This sentence was corrected

221-222 and Table 2: 2+1+37=40 (not 39), 0+1+9=10 (not 9), but these number should be re-calculated basing on new alignments

Author’s: The numbers were corrected based on the new alignment.

242: psbA-trnH should be in italics

Author’s: This sentence was corrected.

244: "indicating little genetic variability intraspecific" this part of sentence should be rewritten

Author’s: This sentence was rewritten.

363: marker name should be in italics

Author’s: This sentence was corrected.

Reviewer 2 Report

The authors made minor revisions to the manuscript, some questions were simply not answered, why?

I noticed that some data were changed in table 2, without a justification from the authors, what is the reason?

The introduction has not changed and continues to emphasize something that has not been studied, why?

Author Response

The authors made minor revisions to the manuscript, some questions were simply not answered, why?

I noticed that some data were changed in table 2, without a justification from the authors, what is the reason?

Author’s: We reviewed the sequences, identify the mistakes and correct them, and justified according to the changes.

The introduction has not changed and continues to emphasize something that has not been studied, why?

Author’s: We attended your suggestion and added a paragraph to the discussion reporting the biological properties of the main compounds of C. verum (Lines 219-226).